# Application of the Social Cognitive Theory amid food parenting practices of Black immigrant mothers in the US: A qualitative study

**Phoebe P. Tchoua** *¤, **Mary Brannock, Deborah Quesenberry**

Department of Community and Behavioral Health, College of Public Health, East Tennessee State University, Johnson City, Tennessee, United States of America

¤ Current address: Department of Health Behavior, Gillings School of Global Health, The University of North Carolina at Chapel Hill, Chapel Hill, North Carolina, United States of America

* ptc@unc.edu

## Abstract

**Data Availability Statement:** All focus group data files are available from the Carolina Digital Repository database (ms35tm58g). https://cdr.lib.unc.edu/concern/data_sets/ms35tm58g

### Background

Children of Black immigrant parents living in the US are at elevated risk of being overweight or obese, thus increasing their risks of morbidity and mortality as they age. Parents play a crucial role in shaping their children's nutrition through their food parenting practices. The Social Cognitive Theory (SCT) can explain Black immigrant mother's FPP and their children's dietary behavior. This study aimed to assess SCT's constructs, personal (maternal knowledge, attitudes, beliefs) and environmental factors (acculturation) in relation to the behavioral factor (food parenting practices) among a sample of Black immigrant mothers living in Metro Atlanta, Georgia.

### Methods

Convenience sampling was employed to recruit 30 Black immigrant mothers who lived in seven Metro Atlanta, Georgia counties in the summer of 2022. Four focus group interviews were conducted over two weeks. The qualitative data analysis was thematic.

### Results

Focus group data analysis revealed seven major themes: knowledge, attitude, belief, modeling, acculturation, coercive control, and structure, and six subthemes. Mothers discussed being intentional about encouraging healthy foods and limiting unhealthy foods for their children. Overall, acculturation influenced mothers' food parenting practices. Since migrating to the US, some mothers' nutrition changed in positive (e.g., eating more fruits) and negative ways (e.g., snacking more) because of schedules, cost, and access. Children ate a mixed diet, the mother's native diet and the American diet, and the former was considered healthier and affordable by most.

**Funding:** This research was supported by a grant from the T32 Cancer Health Disparities Training Grant from the National Cancer Institute of the National Institutes of Health (T32CA128582). The funder had no role in study design, data collection and analysis, or preparation of the manuscript.

**Competing interests:** The authors have declared that no competing interests exist.

## Conclusion

This is the first study to look at the food parenting practices of Black immigrants in the US. By identifying key factors that influence the food parenting practices of this population and their children's dietary habits, this study's findings are useful to practitioners or researchers who work with this population on nutrition.

## Introduction

According to the 2019–2020 National Survey of Children's Health 10- to 17-year-olds with a foreign-born parent had higher BMIs when compared to children whose parents were born in the US [1]. Since children of immigrant parents have an increased risk for being overweight [2], it is important to learn about factors that can contribute to this risk, such as food parenting practices [3]. Research has shown that differences exist between food parenting practices of White and racial/ethnic minority groups [4, 5].

Poor nutrition is a major risk factor for chronic diseases [6–10]. On average, the diet quality of children in the US is poor with a Healthy Eating Index score of 58 out of 100, among individuals 2 and older [11]. Studies have also shown that food parenting practices can directly influence the child's eating habits [12] and the risk of a child being overweight or obese [13]. For this study, food parenting practices are defined as parents' behaviors or actions, as primary caregivers, that affect their child's dietary behavior. Parents play a vital role in developing many aspects of their children's lives, including nutrition. Children depend on their parents for the quantity and quality of food that is available in the home (such as meals and snacks), and parents can create (knowingly or unknowingly) an obesogenic environment in the home [14].

The Social Cognitive Theory (SCT), as a whole or in a modified form, has been used by many researchers to explain human health behavior [15]. It is defined as a model of reciprocal causation between personal factors, behavior, and the environment (Fig 1) [16]. When considering the food parenting practices of Black immigrant mothers, the SCT model can be adapted where (1) the personal factors represent the maternal knowledge, attitudes, and beliefs, (2) the behavioral factors are the food parenting practices, and (3) the environment is acculturation, the physical and psychological changes from moving from your native culture into a new culture Fig 2.

Parents' knowledge, attitudes, beliefs, and acculturation towards nutrition influence their food parenting practices and their children's dietary behavior and nutrition. Higher levels of acculturation among family members have been found to be related to an increased incidence of excess body weight in family members, including children and youth [17, 18].

No study, to the authors' knowledge, has looked exclusively at the relationship between food parenting practices of Black immigrant mothers and its influence on their children's nutritional behavior while informed by a theoretical model. Kengneson [19] conducted the first study to identify factors associated with the feeding practices of Black immigrant mothers living in Ottawa, Canada. They concluded by highlighting the need for more qualitative studies to better understand the influence of the feeding practices that Black immigrant mothers on their children's dietary behavior, and weight status.

In this study, we aimed to assess the SCT's constructs among a sample of Black immigrant mothers living in Metro Atlanta, Georgia. Specifically, we wanted to assess maternal knowledge, attitudes, beliefs, acculturation, and modeling in relation to food parenting practices among a sample of participants who previously completed a modified Comprehensive Home

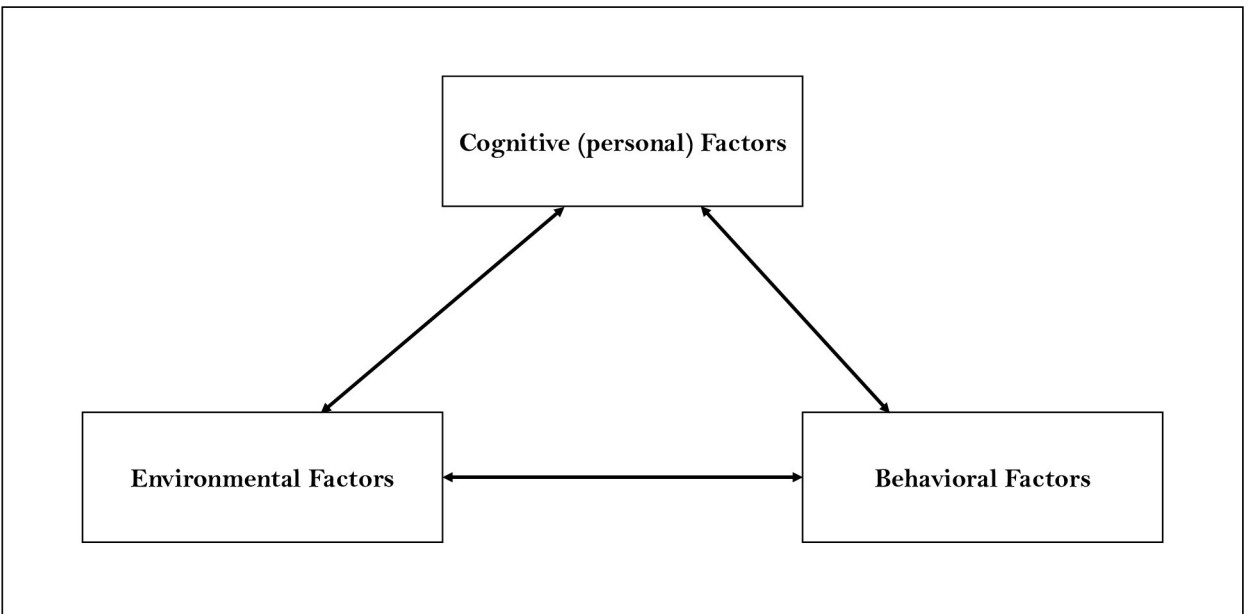

**Fig 1. Social Cognitive Theory model.**

Environment survey using focus groups. This study was exploratory; therefore, the researchers did not have any prior hypotheses.

## Methods

### 2.1. Participants

Four focus groups consisting of 6–12 participants from Metro Atlanta, Georgia were held on *Zoom* (version 5.3.0) due to COVID-19 (N = 30) during August 2022. Initially, focus groups

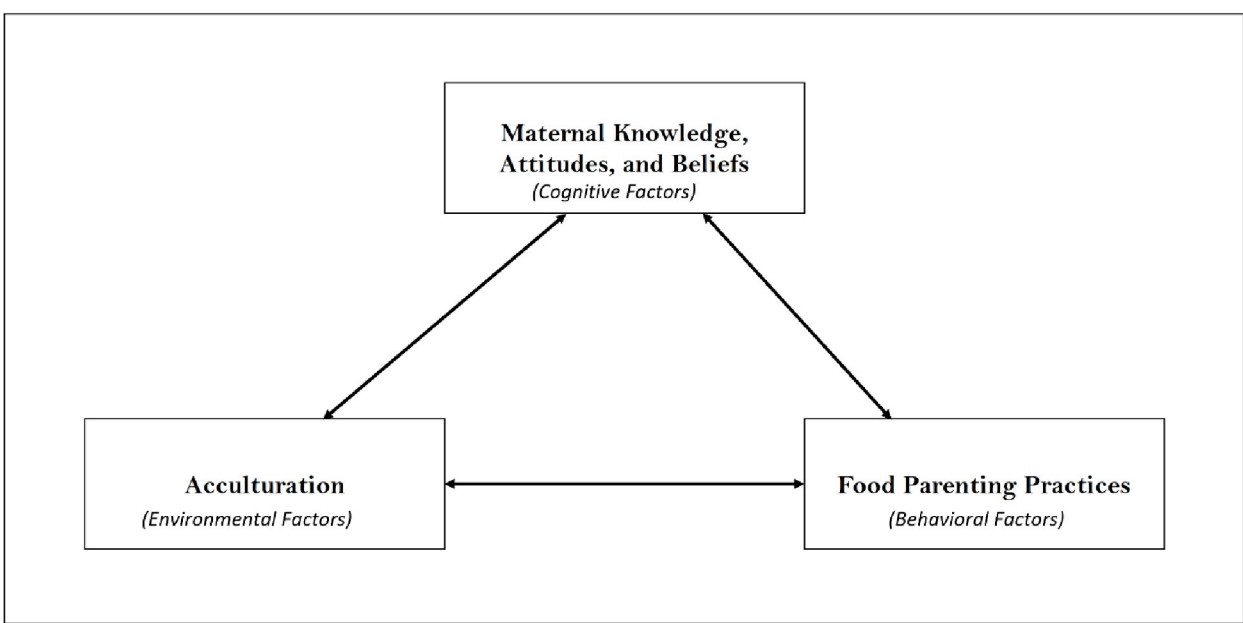

**Fig 2. Adapted Social Cognitive Theory model to include food parenting practices and influences.**

were scheduled based on children's age (e.g., 2–4 years old, 5–9 years old, 10–14 years old, and 15–19 years old). However, due to scheduling conflicts and low sign-ups for each group, only one focus group was based on age. The first focus group included mothers of children aged 2–4 years old. The remaining focus groups included mothers of children aged 2–11 years old. Thirty mothers from seven African and Caribbean countries, Cameroon, Ghana, Kenya, Nigeria, South Africa, Senegal, Trinidad and Tobago, participated in the four focus group discussions. The length of stay in the US for participants ranged from 5 to 25 years.

## 2.2. Recruitment, participant selection

Participants were recruited by PT (the first author) via flyers (Facebook, Reddit, Instagram, and WhatsApp), and emails to relevant community organizations, and snowball sampling from 15, July 2022–23, August 2022. After visiting the study link, found on study flyers, study participants took an 8-question screening survey to determine eligibility, read the consent form, and gave their digital consent to participate in a focus group. Study inclusion criteria were females over 18 years of age, born in an African, Caribbean, or Latin American country, fluent in English, of Black, African, or Afro-Caribbean race or ethnicity, who resided in one of 11 metro Atlanta counties, with a child between the age of 2 and 19, and the primary caregiver of the child. Parents were excluded if they did not meet all eight inclusion criteria. Study participants received a $10 Tango gift card as an incentive.

The study protocol was reviewed and approved by the East Tennessee State University (ETSU) Institutional Review Board. Digital consent, by clicking "I agree: (a) I have read the consent form, (b) I agree to volunteer, and (c) I am physically present in the United States", was gained from each participant prior to study involvement.

## 2.3. Measures

A semi-structured discussion guide that consisted of 15 questions (see Table 1), developed by the principal investigator (PI) and two qualitative research experts, to elicit data on the role of acculturation in the child's dietary behavior, the effects of the mother's eating behavior on her child, how the mother's knowledge and beliefs impact the child's eating behavior, the food parenting practices of the mother, and her learning needs about nutrition. During the data collection process, the guide was re-evaluated to ensure that rich and in-depth data were collected.

## 2.4. Data collection and data analysis

The analytic plan was pre-specified, and any data-driven analyses were clearly identified and discussed appropriately. The data analysis was thematic [20] following the modified SCT constructs and some questions from the focus group semi-structured discussion guide as central themes. Analysis also allowed for the identification of emergent themes beyond those contained within the SCT model. The researchers combined inductive and deductive coding approaches. The focus groups were held on Tuesday evenings and Saturday mornings or early afternoon. Seven countries were represented across all focus groups, with most participants emigrating from African countries. Focus groups lasted 78 (focus group 1), 90 (focus group 2 and 3), and 84 (focus group 4) minutes, were digitally recorded with participants' consent, and the semi-structured interviews continued until no new thematic ideas were generated. Focus group attendance was limited to the participants, the researcher, and research assistants.

The recorded focus group interviews were transcribed verbatim. A codebook was generated through an iterative process whereby themes were generated from the focus group data, SCT constructs, and questions from the focus group interview guide. Inter-rater reliability was assessed through a process where 25% of the data [21] were double-coded using percent

**Table 1. Focus group semi-structured discussion guide.**

| | |
|---|---|
| **To understand how the process of adopting the American culture as a Black immigrant can impact the child's dietary behavior:** | |
| Q1 | Let's begin by having each of you describe an average day feeding your child(ren). |
| | Probe: breakfast, lunch, dinner, snacks |
| Q2 | So, since moving to the US, how has what you eat changed? |
| | Probe: types of food and beverages, the timing of meals, snacking |
| | Probe: What do you eat/drink more of? Less of? |
| Q3 | What about your children? How has what you feed them changed since moving to the US? |
| | (If your child was born in the US, what is the difference between what you feed them here vs. what you fed them in your native country?) |
| Q4 | What things would you say caused these changes? |
| | Probe: work schedule, food availability, childcare, proximity to parents/family members? |
| | Probe: In what ways? |
| Q5 | Among the things you shared, which ones have had the greatest influence on what and how you feed your child? |
| | Probe: Why did these factors have such a significant influence? |
| **To understand how the eating behavior of the mother affect that of children:** | |
| Q6 | Do you use food to deal with boredom, anger, bad mood, to celebrate, or treat yourself? |
| | Probe: If yes, tell me more. |
| | Probe: If yes, what kinds of food? |
| Q7 | If your child was asked to list the foods you normally eat, what foods would they list? |
| Q8 | How do you think about the food you eat in front of your child? |
| | Probe: Do you avoid buying certain foods for your home? |
| Q9 | What differences are there (if any) between what you eat and what your child(ren) eats? |
| **To discover how knowledge and beliefs of the mother impact the child's eating behavior:** | |
| Q10 | What comes to mind when you hear 'US Department of Agriculture dietary recommendation guidelines? |
| Q11 | How do you think the food you feed your children is influenced by what you ate as a child? |
| **To learn about other food parenting practices:** | |
| **Restriction** | |
| Q12 | How do you regulate your child(ren)'s consumption of junk food, such as cookies, chips, sodas, juices, or other processed foods? |
| | Probe: Does your regulation help your child eat fewer processed foods? |
| **Pressure to eat** | |
| Q13 | Do you try to get your child(ren) to eat? |
| | Probe: If yes, describe a typical or recent scenario where this happened. |
| **To learn about food in the school environment** | |
| Q14 | If your child is enrolled in school, do they eat at school or pack their lunch? |
| | Probe: If they eat at school, what are your thoughts on the meals they eat there? |
| | Probe: If they pack their lunch, who typically prepares the lunch and how are decisions made about what to pack? |
| | Probe: Do you think your child(ren) eats better at school or at home? |
| | i. Nutritional quality? |
| | ii. Amount? |
| **The learn about the learning needs of participants:** | |
| Q15 | Before we end, I would like to know what information [and/or resources] about nutrition [including feeding children] you would like to learn more about. |
| | Probe: In what format would you prefer to receive this information? Social media (Facebook, Instagram, TikTok, YouTube), a doctor, health department, or other? |
| | Probe: Any comments in general about what we discussed today? Is there anything I missed? |

agreement among raters as described in McHugh 2012 [22]. Each rater coded the same data independently using a predetermined unique color for each code. Next, one rater counted the number of similar codes and divided that number by the total number of codes to obtain the interrater reliability percent. The initial percentage was less than 80 and both raters met to discuss and resolve discrepancies. Following the discussions, the codebook was modified accordingly to help with consensus building and inter-rater reliability, and percent agreement between raters increased. The team reached 83% coding similarity; and the PI coded the remaining data (i.e., 75%) using the finalized codebook and generated a summary of each theme across the four focus groups. Key focus group findings were shared with participants via an informational graphic, following completion of the research project.

## Results

Thirty mothers over the age of 18 participated in four focus group discussions. The number of participants in each focus group ranged from 6 to 12. The length of stay in the US ranged from 5 to 25 years, and their children ranged in age from 2 to 11 years old. In total, seven major themes (*Knowledge*, *Attitude*, *Beliefs*, *Modeling*, *Acculturation*, *Coercive Control*, and *Structure*) and six subthemes (*Know*, *Want to Know*, *Adopting the US lifestyle (opportunities and challenges)*, *Nutrition Change*, *Child's Diet*: *Common Foods*, and *Child's Diet*: *School Lunch food*) were identified in participants' focus group discussions. Each theme and subtheme are defined below and include quotes from study participants. The author of each quote is identified with an R for respondent and a number. Where R1–R6 are from focus group 1, R7–R16 focus group 2, R17–R24 focus group 3, and R25–R30 focus group 4. All focus group participants are female, and their age is not included to maintain the anonymity of each mother.

### 3.1. Theme 1: *Knowledge*

*Knowledge* consisted of two subthemes, *Know* and *Want to Know*.

**3.1.1. Subtheme 1: *Know*.**   This subtheme is defined as the mother's education, information, and experience on nutrition, and the role of different groups and types of food on health. Across all four focus groups, mothers shared being intentional about limiting or encouraging certain foods. They wanted to limit sugar intake because the child is still developing, candy can cause tooth decay, junk food is not good for the body, and processed food is not as healthy as homecooked meals. As stated here,

"*For me, I avoid [inaudible] and things like candy sweets they're not available in our house like you don't buy them just to stay there, to avoid the too much sugars and maybe the spoiling of teeth.*"–R24

On the other hand, participants shared that eating nutritious foods such as fruits and vegetables, plants, and animal products, was necessary for the child's development. A balanced diet included all food groups, food gave you vitamins, and fruits helped relieve constipation. As shared by this mother,

"*I have um worked on him consuming more fruits than before. He used to be really constipated and I had to do something about it.*"–R23

**3.1.2. Subtheme 2:** *Want to Know.* This subtheme is defined as additional information the mothers would like to receive on nutrition. In three of the focus groups, participants wanted to know which foods boost appetite, keep kids healthy, when to eat certain foods (e.g., time of day), recommended amounts of various macro- and micro-nutrients (e.g., protein, veggies, calcium), how to encourage the child to eat at various stages in childhood, correct serving sizes, and what is truly inside the food served at restaurants. For example, two mothers said,

> "*In my case, I would like to know the foods that my kid should eat in order to stay healthy and grow more healthier and the times to eat . . .*"–R5

> "*It is how to get rid of that snack yeah, it's costly.*"–R4

## 3.2. Theme 2: *Attitude*

This theme is defined as, the mother's accepted way of thinking about food in general and about certain foods, how she ate as a child and its influence on how she feeds her child, her feelings and emotions, her actions, and her tendency to move toward or away from certain foods. In three of the four focus groups, several mothers shared how their childhood dietary habits influenced what they do or do not feed their children. If she liked eating a certain food as a child, she now prepared it for her child (e.g., cooking beans, traditional foods, making home-cooked meals). As stated here,

> "*Yeah, I think feeding is a continuous stuff like a generation stuff. So, seeing our parents fed us with vegetables we as parents now think is ok to continue with the feeding cycle. Like for me I was trained to eat an egg every day, so I also make my baby to do that each morning.*"–R9

> "*It is 90% influenced by the foods I ate as a child.*"–R19

For one mother, she does not force her children to eat what she disliked eating growing up. As she said here,

> "*Growing up I hated beans and fufu [pounded meal of starchy root vegetables or corn]. For that reason, I don't force my kids to eat beans*
>
> *or fufu but my mom that lives with me tries to enforce it, particularly eating beans.*"–R17

As for participants' attitude toward sugar, fast food, and processed foods, mothers in all four focus groups are intentional about minimizing or eliminating their consumption. In some households, fast foods are limited to once or at most twice a week, sweets are an occasional treat, hotdogs are not served because of the nitrate, cereal is not served because of the sugar content, concerns that some processed foods are carcinogenic, dilute juice because of the high sugar content, and disinfect lunch meat before eating. For example, two mothers said,

> "*They'll eat um Cheerios. It's the mixture of the honey versus the plain just so that it's not too sweet.*"–R29

> "*Well, I still try to eat a lot of things I ate back home all the ones I can get here . . . but then you kind of like incorporate some of that you know American food here, pizza. I love pizza though, but I try not to eat it so much. I just try to limit it to maybe once in two weeks or something.*"–R14

### 3.3. Theme 3: *Beliefs*

The theme of "Beliefs" encompasses what the mother has accepted as accurate concerning nutrition, food in the US, the USDA (United States Department of Agriculture) Dietary Guidelines and Recommendations, her thoughts about food and nutrition, and how she would feed her child if she lived in her native country. Across all four focus groups, most mothers shared similar beliefs on what they considered healthy and nutritious for themselves and their children. Some mothers believe fresh and healthy food is not easily accessible in the US due to cost and availability. In some focus groups, mothers stated that eating on the go was not good, and junk food is not good for their system (i.e., overall health). Other perceptions shared by participants included: school lunch and restaurant food are not very nutritious, fast food makes you heavy, candy will cause cavities, fruits help with constipation, large portion size will make you fat, children need to eat less sugar and more body-building food to grow and develop. As stated here:

"...and now I'm actually starting to be conscious about okay this is one portion size. You can't be eating like five people's portion of rice for lunch, and then wonder why you're getting fat."–R18

"Yeah, my middle son, he expects snacks every day which is not good."–R13

Although most mothers shared how difficult it was to eat healthy in the US, one mother had a different experience. She shared that it is possible and easy to stay healthy in the US. She said,

" ...When you want to stay healthy here it's possible and it's very easy. Those things are already arranged in the aisle, they are already there for you to grab."–R11

In three focus groups, when asked to share thoughts that came to mind when they heard about the US Department of Agriculture's dietary recommendation guidelines, mothers shared positive and negative thoughts about various foods, what to eat and avoid, the disparities between countries, and the trustworthiness of government. As stated here,

"What comes to my mind is a chart that has been divided to determine proteins, carbohydrates, things you need to eat, [and] others you need to avoid yeah things like that"–R4

"Lot of lies"–R16

"Lobbied"–R18

### 3.4. Theme 4: *Modeling*

"Modeling" encompasses the perception of whether the mother intentionally or unintentionally influences her child to eat healthy or unhealthy food. It includes what she eats in front of or away from her child, and what foods she avoids. In two focus groups, when asked what their children would list as the food they eat, they listed, African food (e.g., fufu, chicken, fish, afan soup, okra soup, leafy green vegetables, meat, rice), noodles, cake, and fruits. Some mothers avoided buying 'super processed' (e.g., Little Debbie cakes) food, packaged food, candy, cakes, cookies, sweets, and junk foods. As illustrated here,

"I avoid eating certain foods, especially packaged foods. I eat a lot of fruit and drink tons of water. That's one thing my kids would say. I drink a lot of water."–R19

"*I really avoid buying of cakes, cookies, and sweets.*"–R22

One mother shared not avoiding anything. As stated here,

"*. . .I don't do that, whatever is good for me is good for him so sometimes he will tell me a mommy I want a pizza and when I get the pizza, I'll eat a slice myself.*"–R7

In every focus group, some mothers admitted to not eating certain foods (e.g., snacks, sweets, biscuits [also known as cookies], gum, unhealthy food) in front of their children because they didn't want to share the item or wanted their children to have better eating habits. As this mother said,

"*If I want to like eat my [any] snacks or something sweet, anytime I eat in front of them they end up eating it all. So, I just wait till they're not around and then I eat what I want to eat, but it doesn't influence what I buy for the house. I just regulate when I eat the snacks or what-ever my guilty pleasures and then for dinner I just most times I'm eating something different or and I just give them a little. But they're getting used to the fact that I don't eat what every-one else is eating.*"–R2

For one mother, she always ate healthy in front of her children. As stated below,

"*I always eat healthy in front of my kids.*"–R26

### 3.5. Theme 5: *Acculturation*

Acculturation was the common theme, and it is comprised of four subthemes, *Adopting the US Lifestyle (opportunities and challenges), Nutrition Change, Child's Diet: Common Foods*, and *Child's Diet: School Lunch Food.*

**3.5.1. Subtheme 1:** *Adopting the US lifestyle (opportunities and challenges).* This sub-theme definition focuses on perceptions around adopting a US lifestyle, such that, since mov-ing to the US, negative things mothers said about access to, eating and preparing food, and weight, changes in her diet, positive lifestyle changes, and how differently the child would eat if he/she lived in the mother's native country. Overall, across all four focus groups, mothers made four main observations about food in the US, (1) there is an abundance of food (e.g., more snacks and larger portions ), less structured eating, and more fried foods; (2) the food quality, flavor, taste, and nutritional content are different compared to their native foods; (3) food is not as organic or as fresh in comparison to food in their native country; and (4) food is more expensive. For example, these mothers said,

"*Well, definitely is the whole lot more food in this country that isn't necessarily, you know as organic and fresh as back home. and, obviously, the more organic produce is more expensive here than back home and then definitely a lot more snacks are available here than back home, so I think my. well back home, I probably was eating more of the like heavy Nigerian food, you know, beans and rice and fufu and bigger servings*"–R2

"*In South Africa you can get varieties of food, homemade yes. But when you come to Atlanta and you have to buy a lot of stuff it's quite expensive, you know. Yeah, you may look for the same type of food, but you may not get it, so you have to take another option and it becomes expensive.*"–R5

On the other hand, some mothers have experienced positive lifestyle changes in their nutrition. Some have developed better eating habits, become more aware of alternatives to their native foods, learned about portion sizes, diversified their diet, and eat more salads and fruits. Although many mothers shared that the cost of organic produce in the US is prohibitive, one found eating organic food easy. As this mother put it,

"*Um so for me growing up just like most people from Nigeria said we would have three meals my mom made sure of that uh that has changed definitely since relocating for me I rarely eat breakfast I just don't get hungry now because I don't want to. . . Diet has changed. Growing up I don't remember eating salads. Now I'm a salad kind of girl so I would have like a salad around noon for lunch I definitely love Seafood not much of a meat eater, but I would eat fish all day if you let me and I didn't eat that growing up like we always had like beef and stuff like that chicken. And now I'm not really much of a beef or chicken kind of girl, but I'll take fish any day. I love fruit so I would eat fruit I try to snack on healthy stuff even though I have three kids.*"–R30

In participants' native countries, food was varied, natural, and homegrown, most meals were homecooked, some mothers shared eating heavier meals (e.g., big meal for lunch, smaller meal for dinner), on a structured eating schedule (e.g., cooked and ate 3 meals per day), and at the table. Also, some shared that chips or candy were absent in the home, eating out was on occasion, and snacking was limited. As stated here,

"*. . .When I just got here [the US] was different because I was home [native country] all the time, so it was kind of structured back then. I woke up in the morning, I'll get my normal breakfast, eat my lunch, eat my dinner, three straight meals a day. But then when I got into the system when I started getting busy it got so different.*"–R7

"*. . .All right, so literally my diet hasn't changed too as well. I still eat most of my African food and which I also allow my kids to eat too except that I have had this opportunity to have variety of fruits here in America. So that has really, I incorporated that into my diet. I eat variety of fruits now which we don't really too much have access to in Africa.*"–R26

Mothers did not hesitate to share the differences in their children if they lived in their native country. For example, their dietary habits would include fewer snacks, less juice, no cereal, and rich in natural foods and not canned or processed foods. As stated here,

"So, they definitely would not be having as many snacks as they do now, like you know in school, they give them morning snack, afternoon snack and then, when we pick them up from school, we also give them a snack so they probably will not be eating as many snacks."–R2

"*If she was born back home, I would feed her with traditional porridge from millet compared to now . . .again snacks would not be their favorite meals*"–R1

"*If she was born in my native country, I would feed her with Kenyan foods all through. I don't think I would introduce snacks.*"–R6

"*So, I I think that they would there would be a lot less juice they would not consume cereal at all which they do now because it's not as easily accessible back home and it's very expensive. . .*"–R19

**3.5.2. Subtheme 2: *Nutrition Change*.**    This subtheme is defined as the reasons for change in diet in the US (accessibility, cost, availability, schedule). In two of the four focus groups,

work schedule, cost of foods, access to fruits, access to native foods, cost of organic food, and lack of a support system are some of the reasons mothers gave for the change in their dietary habits in the US. The pace of life in the US is very different compared to that of their native country; it is much faster and more stressful. In their native country, they had help from family members, so they didn't feel the stress of raising a family. These are some of the factors responsible for the change in their new lifestyle such as the number of meals which has decreased from three square meals a day to two and one for others, skipping breakfast, cooking less, and cooking weekly instead of daily. As these mothers said,

> "*So, the eating habit back home was very wonderful. I wish I could adopt that here. and then I don't know but I'm trying. Some days it works, some days it doesn't. I know here everything is junk junk junk. Here, the organic is very expensive so we can't even afford it so we have to go with whatever we can afford yeah.*"–R7

> "*. . . And um I've really been drained by the system um eating much of um chips and chicken and cereals and the rest yeah that's my experience.*"–R10

**3.5.3. Subtheme 3: *Child's Diet: Common Foods.*** This subtheme focuses on reports of what the child eats for breakfast, lunch, dinner, and snacks in the US. Across all focus groups, most children's diets consisted of a mixture of African foods and American foods. For breakfast children eat a wide variety of foods, including oatmeal with fruits, eggs, sausage, bacon, bread and tuna, tea, or peanut butter, meat pie, pastries, samosas, pasta and chicken, and chicken nuggets. During lunch at home, they also ate a mixed diet. Children ate mashed potatoes, mac and cheese, chicken nuggets, pasta, salmon, steak, veggies, ugali, and plantains. The dinner menu was similar to the lunch menu. As for drinks, water was the most common. Children also ate a variety of fruits such as oranges, apples, mangoes, and strawberries. Snacks were very popular in most children's diets, and they varied widely. Children ate fruits, chocolates, cookies, vegetables, yogurt, chips, and popcorn. For example, three mothers said,

> "*My kids, they love to eat fufu and okra soup a lot yeah, they love to eat that a lot. They like to eat jollof rice too, vegetables, fish you know stuff like that, mostly like Nigerian meals. They eat other pizzas stuff like that but those are an occasional basis. They like to eat um yam and sauce with vegetables most of their food comes with vegetables spinach or kale you know and some other Nigerian vegetables, like we call it ugwu. It is pumpkin leaves, yeah. So, most times I try to put vegetables in all their food. The okra soup is made with a lot of vegetables too.*"–R14

> "*In the mornings, they eat either bread and peanut butter and some water and a fruit or they eat oatmeal or cheerios and almond milk with the go go's squeeze, those green apple sauce juice packs.*"–R2

> "*. . .and yes, for breakfast mostly it's usually the egg omelets and sausages and milk.*"–R21

In all four focus groups, while most families ate African foods regularly, for a couple of mothers, their native foods were not as frequent, once a week or when visiting grandparents. As stated here,

> "*I think for my household, we are actually different. I don't do a lot of our traditional food, not as often so my son, he knows it and now I still eat it like if we go to my parents and stuff. But I don't; cook it at home often. I cook lots of the American style food. So, it's completely different to what I grew up from what he's eating now.*"–R3

Eating out was not a common practice in many households. Among mothers who talked about eating out, they did so once or twice a week and usually on the weekend. Another food type parents deferred to the weekend was desserts. As this mother put it,

"*You know we eat the same thing African food yeah and restaurant once in a blue moon. I don't like eating out so I'm so picky, I mean everything is homemade. Most of it. once in a blue moon is if you have to eat in the restaurant, it got to be seafood like . . . Yeah, I really watch their diet.*"–R8

**3.5.4. Subtheme 4:** *Child's Diet: School Lunch food.*   This is defined as whether the child eats school lunch or takes a packed lunch, the reasons the mother gives for the child eating school lunch or packed lunch, the mother's feelings/thoughts/opinions about school lunch, and where the mother thinks the child eats better, at school or home. When it came to school lunch, children ate the school lunch or a packed lunch. Overall, across all four focus groups, mothers did not know how well their children ate at school. In three focus groups, mothers felt that their children ate better (i.e., the nutrition of the food) at home than at school, but one mother felt that her child ate better at school because of the discipline and structure. Mothers who packed lunch did so for various reasons, such as the school lunch did not match their standards at home, it was not as nutritious, it cost too much (e.g., $10), no school lunch was offered, the food was unfamiliar to the child, and the child was not going to eat what was on the menu for the day. Usually, mothers packed food that the child would normally eat at home or sandwiches. For example, two mothers said,

"*The best is to pack her lunch, because I don't really know what the school will feed her with, like I don't trust the feeding in school to be the same with my feeding standards at home.*"–R9

"*Yeah, my son is at school. I take him to daycare. So, I think that he eats better at school than at home because there are disciplines not like at home, where I have to force him to eat.*"–R1

There was some positive feedback on school lunch. A few mothers believed that the quality of the school lunch had improved in recent years and acknowledged how hard it can be to try to cook and feed school children. Because of school lunch, some children started eating more vegetables. One mother shared that her daughter started asking for spinach at home after being introduced to it at school. For another, her children only take fruits as snacks to school because they are pushing for less processed food. As this mother said,

"*Never prior to starting school he would not eat broccoli but now he eats broccoli because he was eating it at school . . .so because of that but I think it's introduced them to veggies. I think they've done better my oldest would eat broccoli now and he never ate it as a kid for me before he started school.*"–R30

## 3.6. Theme 6: Coercive control

This theme is defined as what and how mothers restrict unhealthy food and encourage healthy food consumption in their children. Multiple mothers gave examples of how they limit access and opportunity to unhealthy foods (e.g., processed food, junk food). For some, they eliminate sugar, limit consumption of certain ingredients, or swap out unhealthy foods for healthier ones. As stated below,

"*Because I have issues with eating sugar, so I try to not give her anything sugar. I try to reduce like um foods that have a lot of nitrates like you know the um like the sandwich meat and hot dogs and things like that um. I don't give her those things even though she likes them, and she asks for them but I yeah, I don't give it to her*"–R27

"*I am not big on junk foods. So, I try to replace it with fruits, but let them have it every once in a while.*"–R14

Pressure to eat came up when mothers were asked "*if they ever try to get their child to eat.*" This practice was used to insist that the child eat real food and at times mothers used bribes in the negotiating process. As stated below,

"*Um so my daughter doesn't she doesn't really like to eat like. We have to force her to eat like a real food. She's more of a snack and candy person, and so the way we get her to eat, like her vegetables and things is we say, 'well, you need to make sure you finish your food, and if you finish your food, you get a treat like ice cream or not every day, [but you know], or you get a lollipop or if you finish this; over the weekend I'll take you out to go get something sweet' that we know she likes. So that's kind of how we use food.*"–R2

### 3.7. Theme 7: Structure

This is about a system mothers put in place to influence the child to eat in a certain way, such as rules and limits. In addition to placing limits, some study participants also monitor what the child eats. They keep track or directly ask the child; for example, 'what did you eat at school'? As stated below,

"*Yeah, for my kids. I have to regulate the junk like things like soda, things like biscuits, things like lollipop. Yeah, you know these foods are sugary, so they cause lack of appetite to the kid. So, when you give all things like soda, things like ice cream you find that at the end of the day the child is not able to take things like Ugali or the real food now. This will cause an effect to the health of the kid. Thank you. So, I have to regulate the junk.*"–R1

And they also shared rules they employ to minimize the consumption of desserts, sweets, junk foods, etc. Some mothers use a point system to earn certain foods (e.g., candy, juice), some set a limit on how many snacks the child can have a day, and some leave junk and sweets for the weekend. As stated below,

"*So, what I do to regulate that I tell them one sweet a day so it's either um if you get a juice no ice cream um if you got candy at school, you can't come home and get a juice and if they give you candy at school . . .then the juice box it's not like you get it you have to earn it. Like I'm on a point system. If you do what is right the whole week maybe, you can take a juice box with your meal um so if you don't have points don't come and ask me for a juice box so that also helped.*"–R28

"*For mine I actually have a snack he has his own snack bucket, and he gets to go choose the snack. So, he knows that he gets 2 snacks, and he gets to choose whatever 2 snacks that he wants. So, I try to regulate it by teaching him how do to. You get 2 snacks and then after that that's it on what you pick and then, in the afternoon time we don't do juice and stuff.*"–R3

"*We have a rule, example, soda - do not touch; juice every Wednesday.*"–R17

## Discussion

### 4.1. Summary of main findings

The current study was designed to fill an existing gap in the literature about the food parenting practices of Black immigrant mothers in the US while guided by the SCT. Specifically, the main objective was to assess maternal knowledge, attitudes, beliefs, acculturation, and modeling in relation to food parenting practices. The four focus groups revealed seven major themes and six subthemes. In short, most Black immigrant mothers in this sample shared similar knowledge, attitudes, and beliefs on nutrition. In fact, they are intentional about encouraging the intake of healthy foods (e.g., fruits, vegetables, water, minimally processed foods) and limiting the consumption of unhealthy foods (e.g., candy, junk food, sugar sweetened beverages). Additionally, mothers' childhood dietary habits and native culture influence what they do or do not feed their children. Above all acculturation played a key role in their food parenting practices in positive and negative ways.

**4.1.1. Knowledge and beliefs.** Overall, Black immigrant mothers in this sample shared congruent knowledge, attitudes, and beliefs on nutrition, which in turn was found to directly influence their food parenting practices. Mothers confidently shared which foods the child should eat and why. This finding corroborates what Cook [23] found, that Black immigrant mothers in the UK were knowledgeable of what constitutes a healthy diet: lots of fruits and vegetables, whole, natural, and fresh foods. Participants were also educated on, capable of, and confident in preparing healthy meals for their children. Additionally, mothers in this study frequently reported rules about snacking, (i.e., the number, the type, and the timing of snacks). Also, they intentionally kept some foods out of their child's reach, and ensured the child did not eat too many sweets or their favorite unhealthy foods.

On the other hand, focus group participants also shared what additional information they would like to receive on nutrition, notably foods that can boost appetite in a child that does not like to eat minimally processed food, foods that help the child to grow and remain healthy, and how much to consume from each food group. Most mothers were clear on what foods they considered healthy and unhealthy. A prevailing belief among mothers was that fresh food is not easily accessible in the US, and it is expensive. Many perceived their native foods to be healthier and less processed. These results match those observed in an earlier study by Jakub [24] and a scoping review [25], where some Black African immigrants considered their native foods, including those with high-fat content, healthier compared to American foods because African cuisine is primarily minimally processed, unlike American cuisine.

Although mothers displayed confidence in their knowledge and beliefs on healthy and nutritious foods for their children, they also wanted to obtain more information on this topic. In sum, there are three main points. Mothers seem to be doing the best they know how at this point and with the information they have. Second, they are concerned whether the offered food is indeed the best for their child. Finally, mothers want more information to confirm that what they are doing is indeed the best approach for feeding the child. As previously mentioned, SCT has been used to help explain human health behavior [16, 26]. Focus group discussions confirmed the relationship between the personal and behavioral factors in this group of Black immigrant mothers. Specifically, a mother's nutritional knowledge (personal factors) directly influences her food parenting practices (behavioral factors) and her child's dietary habits.

**4.1.2. Attitudes.** Parent's childhood experiences influence how they feed their children [27]. A mother's perception about food influences how she feeds her children, and these perceptions have been influenced by how her parents fed her as a child. In this investigation, Black immigrant mothers reported being more likely to prepare homecooked meals if their

mothers did so and also to give their children foods they ate and/or enjoyed as children. Additionally, for many mothers, snacking is not a part of their native culture, and they want to limit the amount and type of snacks their children eat. Further exploration of the nuances of maternal attitudes and the impacts of these attitudes on feeding practices would add to the body of research in this topic.

**4.1.3. Modeling.** Modeling was another theme that emerged during focus group data analysis. When we asked mothers about their diet from their child perspective, they listed African foods, American foods, foods from other nationalities, sweets, and fruits. They also shared which foods they avoid or eat away from their children, such as processed, packaged, and junk foods. These findings are similar to those from previous studies in immigrant populations who valued fresh, natural, minimally processed, and native foods, and homecooked meals over canned, processed, and fast foods [25, 28, 29]. Many mothers reported being conscious of eating more healthily in front of their children. Interestingly, mothers made a distinction between the foods they eat in front and away from their children. Perhaps, children are aware of all the types of foods their mothers eat, and their dietary behavior is influenced beyond the modeling mothers strive to parent overtly.

Participants' comments on attitudes and modeling match findings from a systematic review [30] that revealed that adolescents' belief in healthy eating increased when they saw family members eating healthy food. Youths were more motivated to make healthy or unhealthy choices by observing parents or siblings' food choices. Birch [31] also reported that children's dietary behavior is influenced by observing that of their parents. In short, a mother's eating style can explain her child's eating style [32].

**4.1.4. Acculturation.** A major theme across all focus group interviews was acculturation. This is unsurprising, because studies on immigrant population have previously reported on the key role acculturation plays on eating, nutrition, and weight in both positive and negative ways [5, 17, 33, 34]. The mother's reported knowledge and attitudes, the foods she eats, feeds her children, and the why, were all heavily influenced by the culture of her native country. Most participants shared that eating healthy is important. Mothers believe that fresh food is not easily accessible in the US, eating healthy in the US is difficult, and think that certain foods should be avoided. They also shared that the quality, flavor, taste, and nutritional content of food in the US is not the same, unsatisfactory, compared to the food in their native country. Also, food costs more, and eating organic is not as feasible. Consequently, their eating has changed since migrating to the US. A possible solution to overcome food costs, is to purchase healthy food on sale, eat foods based on their harvest season (e.g., fall foods in the fall, winter foods in the winter), plan the food menu for the week in advance, shop at different grocery stores, and utilize online resources to learn how to save money on purchasing nutritious foods.

For some mothers, their eating has improved since migration in the following ways: becoming aware of portion sizes and eating more fruits and salads. As mentioned earlier, their diets now have some American elements, such as snacking, international cuisines, and eating out. These findings further support the idea of bi-dimensional acculturation, where Black immigrant mothers maintain parts of their native culture and adopt parts of the American culture, experienced by African immigrants women in high-income countries who consumed both their native foods and non-native foods, but preferred their native foods, added salt and other spices to the local food to make it tastier [34].

There are multiple reasons for the change in the diet of Black immigrant mothers since migrating to the US, the cost of food, availability of native foods, excess availability of fast foods, schedules, no to minimal support system, and stress levels. These findings support previous research who reported that African immigrant women adopted American dietary habits more easily because of busier schedules, less time to cook their native foods, and easy access to

unhealthy foods [29, 34]. Alsubhi [5] and Berggreen-Clausen [25] also found that because of work schedules, immigrant mothers had less time to cook for their children, and the cost of healthy food was a barrier to eating healthy.

The children of the women in this study were noted by their mothers to eat a mixed diet of African and American foods. In some homes, the diet is primarily African, in others it is largely American. This finding matches a study [35] where the diet of African immigrant children in the US was significantly influenced by their parents' native culture. Overall, their diet consists of whole foods, homemade foods, fruits, processed foods, and snacks. This is consistent with another study [29] of Black immigrant children who reported eating a mixed diet, and positive diet (i.e., fruits, vegetables, milk) changes since migrating to Canada. A popular drink among children is water, which is the preferred beverage according to the United States Department of Agriculture dietary guidelines [36].

Furthermore, for children enrolled in school, mothers reported that they either eat at school or bring a packed lunch. Mothers are uncertain what the child eats at school, or how much they eat. Many mothers say their children eat more nutritious and better-quality food at home compared to school. Moreover, some reported the positive influence school lunch had on their children, such as eating more vegetables.

The results of this study support the utility of the SCT model in food parenting practices research. In particular, a mother's personal factors (knowledge, attitudes, and beliefs) have a reciprocal relationship with her environment (acculturation) and directly influences her behavior (food parenting practices). More research is needed to explore the relationship between mothers' specific food parenting practices, the child's dietary behavior and weight. Further research might investigate what other elements in a mother's environment influence specific food parenting practices such as social determinants of health, and child's age and sex.

## 4.2. Strengths and limitations

To the best of the researcher's knowledge, this was the first qualitative study on maternal knowledge, attitudes, beliefs, acculturation, and modeling in relation to food parenting practices that was conducted with Black immigrant mothers in the US. Therefore, these findings have major implications for Black immigrants living in the US and other developed countries in the West. This study provides vital and useful information when working with Black immigrants, similar to this sample, on nutrition and obesity prevention. This information can be used in the dissemination and implementation of health knowledge that is culturally relevant to Black immigrants. Another strength of this study is the 83% inter-rater reliability, through percent agreement among raters, in data coding which reduced qualitative study bias. The findings in this report are subject to some limitations. First, the interview guide had not been previously validated in a pilot study and tested for comprehensibility. Second, the data were collected in one metro city in the US. Therefore, the findings cannot be generalized to all Black immigrant mothers in the US but may be transferrable to those who live in similar metropolitan areas in the US. Third, social desirability bias, is a common limitation found in focus group data [37]. We minimized this risk by establishing several ground rules (e.g., identities of group members will remain confidential; group members have a right to their viewpoints and opinions) at the beginning of each focus group. Nevertheless, this research extends our knowledge on how Black immigrant mothers in the US feed their children and the many factors that influence those decisions.

## Conclusion

In sum, this qualitative study provides important information on the unique perspective and challenges of food parenting practices of Black immigrant mothers in the US. The focus group discussions revealed this study population has a unique challenge in feeding their children in the US. Their personal factors (i.e., knowledge, attitude, belief) and environmental factors (i.e., acculturation) may directly influence their behavioral factors (i.e., food parenting practices) which in turn may impact their children's dietary behavior. This study highlights essential areas to consider when developing nutritional or diet quality interventions for this population.

## Acknowledgments

Shahin Zare, MPH and Saudi Mamudu, MPH assisted in the focus group data collection.

## Author Contributions

**Conceptualization:** Phoebe P. Tchoua.

**Data curation:** Phoebe P. Tchoua.

**Formal analysis:** Phoebe P. Tchoua, Mary Brannock.

**Investigation:** Phoebe P. Tchoua.

**Methodology:** Phoebe P. Tchoua.

**Project administration:** Phoebe P. Tchoua.

**Resources:** Phoebe P. Tchoua.

**Software:** Phoebe P. Tchoua.

**Supervision:** Deborah Quesenberry.

**Writing – original draft:** Phoebe P. Tchoua.

**Writing – review & editing:** Phoebe P. Tchoua, Mary Brannock, Deborah Quesenberry.

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
