## [Decision Letter · Decision Letter 0]

2 Jun 2024

PONE-D-24-03704Application of the Social Cognitive Theory amid food parenting practices of Black immigrant mothers in the US: A qualitative studyPLOS ONE

Dear Dr. Tchoua,

Thank you for submitting your manuscript to PLOS ONE. After careful consideration, we feel that it has merit but does not fully meet PLOS ONE’s publication criteria as it currently stands. Therefore, we invite you to submit a revised version of the manuscript that addresses the points raised during the review process.

Dear editors.

After evaluation by the reviewers, my advice is to make further revisions to the manuscript.

Thank you in advance.

We look forward to receiving your revised manuscript.

Kind regards,

Thales Philipe Rodrigues da Silva, Ph.D

Academic Editor

PLOS ONE

Journal Requirements:

2. Thank you for stating the following financial disclosure: "This research was supported by a grant from the T32 Cancer Health Disparities Training Grant from the National Cancer Institute of the National Institutes of Health (T32CA128582). "

Additional Editor Comments:

Dear editors.

After evaluation by the reviewers, my advice is to make further revisions to the manuscript.

Thank you in advance.

Reviewers' comments:

Reviewer's Responses to Questions

**Comments to the Author**

1. Is the manuscript technically sound, and do the data support the conclusions?

Reviewer #1: Partly

Reviewer #2: Partly

2. Has the statistical analysis been performed appropriately and rigorously? 

Reviewer #1: I Don't Know

Reviewer #2: Yes

3. Have the authors made all data underlying the findings in their manuscript fully available?

Reviewer #1: Yes

Reviewer #2: Yes

4. Is the manuscript presented in an intelligible fashion and written in standard English?

Reviewer #1: Yes

Reviewer #2: Yes

5. Review Comments to the Author

Reviewer #1: TO THE AUTHORS

SUMMARY This study evaluated Social Cognitive Theory constructs, personal (maternal knowledge, attitudes, and beliefs), and environmental factors (acculturation) in relation to food parenting practices in Black immigrant mothers residing in the US. Data analysis revealed that knowledge, attitude, belief, modeling, acculturation, coercive control, and structure, influenced food parenting practices and their children’s eating habits. Children consumed both the mother’s native diet and the American diet. Mothers encouraged healthy foods by eating more fruits and limiting unhealthy foods for their children, although children snacked more due to work schedules, food costs and accessibility. In conclusion, this study identified the food parenting practices of Black immigrant women associated with the dietary habits of their children.

Indeed, there is a gap in the literature of studies focusing on the dietary behavior of migrant families residing in the US. Given that the US is a multicultural society, from a public health perspective, this topic would be of interest to health professionals working with immigrants in the US. Overall, a good effort was made by the authors. However, there are areas that require clarification that will improve study comprehensibility, transparency and replication. Please refer to my comments to the authors

COMMENTS TO THE AUTHORS

MAIN MANUSCRIPT

Introduction

-line 83, add references to the few studies

Methods

-Line 101, 'All participants providThe first focus group included mothers of children aged 2-4 years old.

Revise is not comprehensible.

-Line 103, 'Thirty mothers from seven African and Caribbean countries participated in the four focus group discussions. ' List the 7 countries.

- Lines 121-127 can be deleted 'All four focus groups were moderated by PT...............,The research team (PT, SZ, and SM) facilitated the sessions.

-In the methods section, assessment tools should be described, namely the questionnaire. The role and contribution of each co-author should be mentioned in the Author's contribution section. Any researchers who assisted in data collection and who are not co-authors should be mentioned in the acknowledgement section of the paper (i.e., SZ, SM).

-Was this questionnaire validated in a pilot study and tested for comprehensibility?

If yes, add a reference to this study or a protocol study. If not, mention this in the study limitations.

Section 2.3 Measures should be revised.

-Line 137 'The hypothesis was specified before the data were collected'.

State the hypothesis under investigation on line 94 of the introduction, and delete line 137 from Section 2.4 Data Collection.

-Lines 143-153 ‘A codebook was generated through an iterative process whereby themes were generated from the focus group data. Inter-rater reliability was assessed through a process where 25% of the data were double-coded by PT and MB (the second author, and a doctoral student experienced in qualitative data ……….following completion of the research project.’

-Line 146 delete author initials. 'by PT and MB (the second author, and a doctoral student experienced in qualitative data analysis'

-line 148-149, Differences in codes were discussed, and resolved, and the codebook was modified accordingly to help with consensus building and inter-rater reliability.

Was a statistical test used to assess inter-rater reliability, e.g. Cohen's Kappa?

-Line 150 ‘reached 83% coding similarity;

How was 83% coding similarity calculated? Mention this in text.

Results

-Line 155, ‘During two weeks in August 2022.’ Redundant, delete from text.

Study duration was mentioned on line 98 of methods.

- Lines 157-159 ‘The focus groups were held on Tuesday evenings and Saturday mornings or early afternoon. Seven countries were represented across all focus groups, with most participants emigrating from African countries’.

Move to methods under the subheading of data collection/data analysis.

- Line 183, delete 'body-building foods' not appropriate word, replace by nutritious foods

- Line 223, define in brackets 'fufu' (legume?)

Discussion

-lines 564-565 'This finding agrees with [21] results, nutritional belief was significantly associated with feeding style. Also, mothers confidently shared which foods the child should eat and why’

Ambiguous. Elaborate by outlining the study by Boucher et al (ref 21)

- Line 573-576 , 'On the other hand, participants also shared what additional information they would like to ............to consume from each food group.’

Are you referring to the present study or ref 21? It is not clear from this statement. Revise

- Line 588 'As previously SCT has been used to help explain human behaviour'.

Add a reference to support this statement.

- Line 677. ‘Another strength of this study is the 83% inter-rater reliability in data coding which reduced qualitative study bias.’

In data-analysis, mention which statistical test was used to derive 83% inter-rater reliability.

-Line 682 ‘We minimized this risk by establishing ground rules at the beginning of each focus group’

What do you mean by ground rules? Inclusion criteria?

Conclusion

The conclusion should concisely summarize the main findings.

-line 689-691, ‘To the authors’ knowledge, this is the first study in the US to explore the food parenting practices of Black immigrant mothers while informed by the SCT’

Delete. This statement was mentioned on line 670 of the discussion

- Line 695 ’The child’s dietary behavior, food choice, and eating behavior can be predictive of adiposity and the attendant morbidity that accompanies excess body weight [36, 37].’

Delete this statement. High BMI was not evaluated in this study.

-Line 697 ‘This study highlights essential areas to consider when developing nutritional or obesity interventions for this population.’

Revise, for the same reason as above.

References

Ref 7 UNICEF. Add link to the website.

Reviewer #2: This is an excellent and well-written paper that explored a well-known and topical issue but in an uncharted subpopulation in the US. The study aimed to use the Social Cognitive Theory (SCT) framework to assess the constructs of maternal knowledge, attitudes, beliefs, acculturation, and modeling in relation to food parenting practices among a sample of Black immigrant mothers living in Metro Atlanta, Georgia.

The sought to identify and explore these defined SCT constructs as key factors that influence the food parenting practices (FPP) of this population in relation to their children’s dietary habits.

In the abstract, the authors' conclusion implied that the study identified "key factors that influence food parenting practices" and as well "children's dietary habits". However the lowest end in the pathway of their SCT construct stopped at FPP. This calls for a delicate balancing of what could or not be concluded by this study.

The authors should reconsider their statement under data analysis that "hypothesis was specified" suggesting that the attempt was to test a hypothesis, which is outside the remit of qualitative research. Qualitative research 'explains' the how or why a phenomenon occurs, and not necessary 'establish' relationship.

The authors did very well in organizing the 'predefined' and emergent themes and subthemes. However, one could only imply that perhaps both closed and open coding approaches were employed; this should be explicitly stated. That way, it would be clear that some subthemes, for instance, "know" which apparently has the same presumed definition as the theme "knowledge", had been predefined, and is clearly one of the preset SCT constructs that the study aimed to assess.

The authors could also consider a further analysis of the theme, "attitude", as it appears that there are subthemes that may bother on the presence versus absence of self-conflicts. There is a suggestions that some of the mothers themselves may be conflicted in what emerged as the attitudes that they displayed.

On a trivial note, the numbering of the themes and subthemes could be revisited; for instance, it is not clear why the first theme was numbered 3.1 and not simply 1, and the subthemes 1.1 and 1.2 in tow.

Similarly, the Editor should advise on if or what any numbering style should feature in the Discussion section. That said, authors should note that the subheading titled, 'summary of main findings' under the Discussion, does not do justice to the depth of all that was presented here.

In addition, authors should avoid repeating the participants' verbatim quotes (i.e., results) in the Discussion.

In the conclusion, the authors should consider revising the use of these phrases: "directly influence" and "in turn impact" given that the qualitative research approach or the findings of this study would or did not establish such level of inferences.

Also it is not typical to make conclusions by citing other researchers' work or statements [Lines 695 & 696]; given that the study did not explore child dietary behaviour or body weight, such statements could be avoided as a conclusion of this study.

Editor should double-check lines 56, 101, and 555 for editorial issues.

6. PLOS authors have the option to publish the peer review history of their article (what does this mean?). If published, this will include your full peer review and any attached files.

Reviewer #1: No

Reviewer #2: **Yes: **Seye Babatunde

---

## [Author Response · Author response to Decision Letter 0]

2 Jul 2024

Reviewer 1

RESPONSE: The manuscript has been modified to meet the PLOS ONE style requirements.

2. Thank you for stating the following financial disclosure: "This research was supported by a grant from the T32 Cancer Health Disparities Training Grant from the National Cancer Institute of the National Institutes of Health (T32CA128582). "

RESPONSE: The funder had no role in the preparation of the manuscript. The edited version of the financial disclosure now reads: "The funders had no role in study design, data collection and analysis, or preparation of the manuscript."

RESPONSE: The ethics statement has been modified as requested. Please see lines 118-121: ” The study protocol was reviewed and approved by the East Tennessee State University (ETSU) Institutional Review Board. Digital consent, by clicking “I agree: (a) I have read the consent form, (b) I agree to volunteer, and (c) I am physically present in the United States”, was gained from each participant prior to study involvement.”

Reviewer's Responses to Questions

Comments to the Author

1. Is the manuscript technically sound, and do the data support the conclusions?

Reviewer #1: Partly

Reviewer #2: Partly

2. Has the statistical analysis been performed appropriately and rigorously?

Reviewer #1: I Don't Know

Reviewer #2: Yes

3. Have the authors made all data underlying the findings in their manuscript fully available?

Reviewer #1: Yes

Reviewer #2: Yes

4. Is the manuscript presented in an intelligible fashion and written in standard English?

Reviewer #1: Yes

Reviewer #2: Yes

5. Review Comments to the Author

 

Reviewer #1: TO THE AUTHORS

SUMMARY This study evaluated Social Cognitive Theory constructs, personal (maternal knowledge, attitudes, and beliefs), and environmental factors (acculturation) in relation to food parenting practices in Black immigrant mothers residing in the US. Data analysis revealed that knowledge, attitude, belief, modeling, acculturation, coercive control, and structure, influenced food parenting practices and their children’s eating habits. Children consumed both the mother’s native diet and the American diet. Mothers encouraged healthy foods by eating more fruits and limiting unhealthy foods for their children, although children snacked more due to work schedules, food costs and accessibility. In conclusion, this study identified the food parenting practices of Black immigrant women associated with the dietary habits of their children.

Indeed, there is a gap in the literature of studies focusing on the dietary behavior of migrant families residing in the US. Given that the US is a multicultural society, from a public health perspective, this topic would be of interest to health professionals working with immigrants in the US. Overall, a good effort was made by the authors. However, there are areas that require clarification that will improve study comprehensibility, transparency and replication. Please refer to my comments to the authors

COMMENTS TO THE AUTHORS

MAIN MANUSCRIPT

Introduction

1. -line 83, add references to the few studies

RESPONSE: Thank you for pointing it out. We did not find any study in the US for Black immigrant mothers. We deleted the first sentence (line 82)

Methods

2. -Line 101, 'All participants providThe first focus group included mothers of children aged 2-4 years old.

Revise is not comprehensible.

RESPONSE: Thank you for pointing it out. We revised the text. Deleted “'All participants provid” (line102)

3. Line 103, 'Thirty mothers from seven African and Caribbean countries participated in the four focus group discussions. ' List the 7 countries.

RESPONSE: Thank you for pointing it out. We added the name of the 7 countries. (Line104-105)

Line 104 – 106: “Thirty mothers from seven African and Caribbean countries, Cameroon, Ghana, Kenya, Nigeria, South Africa, Senegal, Trinidad and Tobago, participated in the four focus group discussions.”

4. Lines 121-127 can be deleted 'All four focus groups were moderated by PT...............,The research team (PT, SZ, and SM) facilitated the sessions.

RESPONSE: We deleted those lines. (lines 123-130)

5. In the methods section, assessment tools should be described, namely the questionnaire. The role and contribution of each co-author should be mentioned in the Author's contribution section. Any researchers who assisted in data collection and who are not co-authors should be mentioned in the acknowledgement section of the paper (i.e., SZ, SM).

RESPONSE: The measure section includes the type of questionnaire used and it is also in Table 1. We removed the research assistant initials. 

6. Was this questionnaire validated in a pilot study and tested for comprehensibility?

If yes, add a reference to this study or a protocol study. If not, mention this in the study limitations.

RESPONSE: Thank you for pointing it out. No, the questionnaire was not validated in a pilot study. It was reviewed after each focus group. We added this as a limitation in the limitation section (line 689-690)

Line 689-690: “First, the interview guide had not been previously validated in a pilot study and tested for comprehensibility.”

Section 2.3 Measures should be revised.

7. Line 137 'The hypothesis was specified before the data were collected'.

State the hypothesis under investigation on line 94 of the introduction, and delete line 137 from Section 2.4 Data Collection.

RESPONSE: Thank you for pointing it out. We added a sentence in the introduction section (line 94-95) about the hypothesis. “This study was exploratory; therefore, the researchers did not have any prior hypotheses.”

Hypothesis information in section 2.4 was deleted (line 140)

8. Lines 143-153 ‘A codebook was generated through an iterative process whereby themes were generated from the focus group data. Inter-rater reliability was assessed through a process where 25% of the data were double-coded by PT and MB (the second author, and a doctoral student experienced in qualitative data ……….following completion of the research project.’

Line 146 delete author initials. 'by PT and MB (the second author, and a doctoral student experienced in qualitative data analysis'

RESPONSE: We deleted the initials. (lines 161, 165, 1667)

9. line 148-149, Differences in codes were discussed, and resolved, and the codebook was modified accordingly to help with consensus building and inter-rater reliability.

Was a statistical test used to assess inter-rater reliability, e.g. Cohen's Kappa?

RESPONSE: Thank you for pointing it out. We used percent agreement. 

Line 155-157: “Inter-rater reliability was assessed through a process where 25% of the data [21] were double-coded using percent agreement among raters as described in McHugh 2012 [22].”

10. Line 150 ‘reached 83% coding similarity;

How was 83% coding similarity calculated? Mention this in text.

RESPONSE: We counted the number of similar codes and divided that number by the total number of codes to obtain the interrater reliability percent. The text was edited to clearly state how we calculated the coding similarity.

Line 155-166: “Inter-rater reliability was assessed through a process where 25% of the data [21] were double-coded using percent agreement among raters as described in McHugh 2012 [22]. Each rater coded the same data independently using a predetermined unique color for each code. Next, one rater counted the number of similar codes and divided that number by the total number of codes to obtain the interrater reliability percent. The initial percentage was less than 80 and both raters met to discuss and resolve discrepancies. Following the discussions, the codebook was modified accordingly to help with consensus building and inter-rater reliability, and percent agreement between raters increased. The team reached 83% coding similarity;”

Results

11. Line 155, ‘During two weeks in August 2022.’ Redundant, delete from text.

Study duration was mentioned on line 98 of methods.

RESPONSE: Thank you for pointing it out. We deleted that part of the text. (line 171)

12. Lines 157-159 ‘The focus groups were held on Tuesday evenings and Saturday mornings or early afternoon. Seven countries were represented across all focus groups, with most participants emigrating from African countries’.

Move to methods under the subheading of data collection/data analysis.

RESPONSE: Thank you for pointing it out. We moved lines 173-1175 to line 146-148.

14. Line 183, delete 'body-building foods' not appropriate word, replace by nutritious foods

RESPONSE: ‘body-building’ was a word used by a study participant. The word has been replaced with ‘nutritious foods’ as suggested. 

Line 199: “On the other hand, participants shared that eating nutritious foods such as fruits and …”

15. Line 223, define in brackets 'fufu' (legume?)

RESPONSE: Thank you for pointing it out. We added a definition in brackets.

Line 239: ““Growing up I hated beans and fufu [pounded meal of starchy root vegetables or corn].”

Discussion

16. lines 564-565 'This finding agrees with [21] results, nutritional belief was significantly associated with feeding style. Also, mothers confidently shared which foods the child should eat and why’

Ambiguous. Elaborate by outlining the study by Boucher et al (ref 21)

RESPONSE: Thank you for pointing it out. We deleted that sentence. (Lines 581-582)

17. Line 573-576 , 'On the other hand, participants also shared what additional information they would like to ............to consume from each food group.’

Are you referring to the present study or ref 21? It is not clear from this statement. Revise

RESPONSE: We are referring to the present study. We added “…focus group participants to make that clearer”

Line 590: “On the other hand, focus group participants also shared what additional information they…”

18. Line 588 'As previously SCT has been used to help explain human behaviour'.

Add a reference to support this statement.

RESPONSE: We added 2 references.

Line 606: “As previously mentioned, SCT has been used to help explain human health behavior [16, 26].”

19. Line 677. ‘Another strength of this study is the 83% inter-rater reliability in data coding which reduced qualitative study bias.’

In data-analysis, mention which statistical test was used to derive 83% inter-rater reliability.

RESPONSE: We added the method used to reach 83% inter-rater reliability, “percent agreement”.

Lines 696-697: “Another strength of this study is the 83% inter-rater reliability, through percent agreement among raters, in data coding which reduced qualitative study bias.”

20. Line 682 ‘We minimized this risk by establishing ground rules at the beginning of each focus group’

What do you mean by ground rules? Inclusion criteria?

RESPONSE: Thank you for pointing it out. We added examples of the ground rules in parentheses to make that statement clearer.

Lines 703 – 705: “We minimized this risk by establishing several ground rules (e.g., identities of group members will remain confidential; group members have a right to their viewpoints and opinions) at the beginning of each focus group.”

Conclusion

The conclusion should concisely summarize the main findings.

21. line 689-691, ‘To the authors’ knowledge, this is the first study in the US to explore the food parenting practices of Black immigrant mothers while informed by the SCT’

Delete. This statement was mentioned on line 670 of the discussion

RESPONSE: We deleted that statement. (Lines 711-713)

22. Line 695 ’The child’s dietary behavior, food choice, and eating behavior can be predictive of adiposity and the attendant morbidity that accompanies excess body weight [36, 37].’

Delete this statement. High BMI was not evaluated in this study.

RESPONSE: Thank you for pointing it out. We deleted the entire sentence (Lines 717 – 718)

23. Line 697 ‘This study highlights essential areas to consider when developing nutritional or obesity interventions for this population.’

Revise, for the same reason as above.

RESPONSE: We deleted the word ‘obesity’. (Line 719)

Lines 718-7120: “This study highlights essential areas to consider when developing nutritional or diet quality interventions for this population.”

References

24. Ref 7 UNICEF. Add link to the website.

RESPONSE: Thank you for pointing it out. We added link to website. 

 [7] U.N.I.C.E.F. Early Childhood Nutrition, https://www.unicef.org/nutrition/early-childhood-nutrition (accessed May 27, 2024).

Reviewer #2: This is an excellent and well-written paper that explored a well-known and topical issue but in an uncharted subpopulation in the US. The study aimed to use the Social Cognitive Theory (SCT) framework to assess the constructs of maternal knowledge, attitudes, beliefs, acculturation, and modeling in relation to food parenting practices among a sample of Black immigrant mothers living in Metro Atlanta, Georgia.

The sought to Identify and explore these defined SCT constructs as key factors that influence the food parenting practices (FPP) of this population in relation to their children’s dietary habits.

1. In the abstract, the author’' conclusion implied that the study identified“"key factors that influence food parenting practice”" and as well“"childre’'s dietary habit”". However the lowest end in the pathway of their SCT construct stopped at FPP. This calls for a delicate balancing of what could or not be concluded by this study.

RESPONSE: Thank you for pointing it out. “Obesity prevention” was removed from the abstract conclusion (line 45

---

## [Decision Letter · Decision Letter 1]

24 Jul 2024

Application of the Social Cognitive Theory amid food parenting practices of Black immigrant mothers in the US: A qualitative study

PONE-D-24-03704R1

Dear Dr. Tchoua,

We’re pleased to inform you that your manuscript has been judged scientifically suitable for publication and will be formally accepted for publication once it meets all outstanding technical requirements.

Kind regards,

Thales Philipe Rodrigues da Silva, Ph.D

Academic Editor

PLOS ONE

Additional Editor Comments (optional):

Dear editor,

I am pleased to inform you that manuscript has been accepted for publication. 

My comments, and any additional reviewer comments, can be found below.

Thanks to the authors for addressing all of the reviewer comments, I have accepted the paper for publication.

The only modification required will be to improve the quality of the image resolution.

Kind regards

Reviewers' comments:

Reviewer's Responses to Questions

**Comments to the Author**

1. If the authors have adequately addressed your comments raised in a previous round of review and you feel that this manuscript is now acceptable for publication, you may indicate that here to bypass the “Comments to the Author” section, enter your conflict of interest statement in the “Confidential to Editor” section, and submit your "Accept" recommendation.

Reviewer #1: All comments have been addressed

Reviewer #2: All comments have been addressed

2. Is the manuscript technically sound, and do the data support the conclusions?

Reviewer #1: Yes

Reviewer #2: Yes

3. Has the statistical analysis been performed appropriately and rigorously? 

Reviewer #1: Yes

Reviewer #2: Yes

4. Have the authors made all data underlying the findings in their manuscript fully available?

Reviewer #1: Yes

Reviewer #2: Yes

5. Is the manuscript presented in an intelligible fashion and written in standard English?

Reviewer #1: Yes

Reviewer #2: Yes

6. Review Comments to the Author

Reviewer #1: The authors have satisfactorily responded to all of my queries.

I have no further comments.

Good luck with future submissions.

Reviewer #2: The authors have adequately addressed all the initial comments. The only addition is a clarification on the initial comment that prompted the authors' response on welcoming suggestions to enhance the sub-section titled "Summary of Findings" under Discussion section. To rephrase, the initial comment was a query on the use of that subheading. This is because a Discussion section is not merely a 'summary' of findings, it is more than that; it provides an 'interpretation' of findings and in the context of what is already known about the topic.

7. PLOS authors have the option to publish the peer review history of their article (what does this mean?). If published, this will include your full peer review and any attached files.

Reviewer #1: No

Reviewer #2: **Yes: **SEYE BABATUNDE

---

## [Editor Report · Acceptance letter]

29 Jul 2024

PONE-D-24-03704R1 

PLOS ONE

Dear Dr. Tchoua, 

I'm pleased to inform you that your manuscript has been deemed suitable for publication in PLOS ONE. Congratulations! Your manuscript is now being handed over to our production team.

Kind regards, 

on behalf of

Dr. Thales Philipe Rodrigues da Silva 

Academic Editor

PLOS ONE